# Does resistance to style-transfer equal Shape Bias? Evaluating shape bias by distorted shape

## Abstract

Deep learning models are known to exhibit a strong texture bias, while human tends to rely heavily on global shape for object recognition. The current benchmark for evaluating a model's shape bias is a set of style-transferred images with the assumption that resistance to the attack of style transfer is related to the development of shape sensitivity in the model. In this work, we show that networks trained with style-transfer images indeed learn to ignore style, but its shape bias arises primarily from local shapes. We provide a **Distorted Shape Testbench (DiST)** as an alternative measurement of global shape sensitivity. Our test includes 2400 original images from ImageNet-1K, each of which is accompanied by two images with the global shapes of the original image distorted while preserving its texture via the texture synthesis program. We found that (1) models that performed well on the previous shape bias evaluation do not fare well in the proposed DiST; (2) the widely adopted ViT models do not show significant advantages over Convolutional Neural Networks (CNNs) on this benchmark despite that ViTs rank higher on the previous shape bias tests. (3) training with DiST images bridges the significant gap between human and existing SOTA models' performance while preserving the model's accuracy on standard image classification tasks; training with DiST images and style-transferred images are complementary, and can be combined to train network together to enhance both the global and local shape sensitivity of the network. Our code will be host in the anonymous github: https://anonymous.4open.science/r/ICLR2024-DiST/

## 1 Introduction

Deep learning models for object recognition are known to exhibit strong texture bias (Geirhos et al., 2018; Baker & Elder, 2022). In solving problems, neural networks tend to discover easy shortcuts that might not generalize well (Ilyas et al., 2019; Drenkow et al., 2021). Rather than learning a more structured representation of objects, such as surfaces and shapes, convolutional neural networks trained for classifying objects rely primarily on the statistical regularities of features discovered along the network hierarchy. Standard networks fumbled badly when the test images were subjected to style or texture transfer (Geirhos et al., 2018), revealing their reliance on texture and local feature statistics, perhaps the easiest features, rather than global shape. Humans, on the other hand, are fairly robust against such style transfer manipulation in object recognition, indicative of our explicit utilization of shapes in object recognition (Ayzenberg & Behrmann, 2022; Ayzenberg & Lourenco, 2022; Quinn et al., 2001a;b).

To close this gap, various approaches have been developed to steer neural network learning towards shapes from texture (Brochu, 2019; Geirhos et al., 2021; Li et al., 2023). Among which, the most effective approach still remains to be augmenting the training data with randomized style-transfer operation (Geirhos et al., 2018; 2021). However, the nature of shape sensitivity promoted by the style-transfer augmented training, and the generalizability of this shape bias to other approaches for evaluating shape sensitivity remain uncertain.

In this paper, we re-examine the role of style-augmentation training and ask whether resistance to style-transfer really reflects models' ability to avoid texture based recognition short-cut. In Figure 1,

**(a) Feature Attribution Results**   **(b) Our Distorted Shape Testbench**

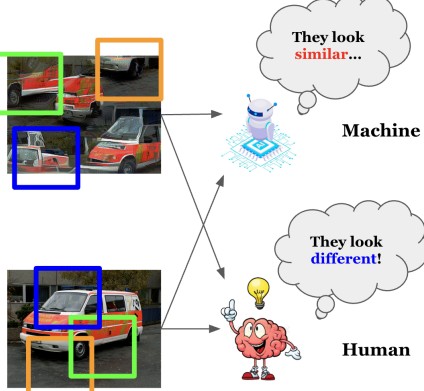

Stylize Augmented Models
still focus on **local feature**

DiST directly shows models'
sensitivity of **global shape**

Figure 1: **Left**: Feature Attribution Analysis based SmoothGrad (Smilkov et al., 2017) on stylized augmentation trained models. Surprisingly, models that can resist style transfers still be primarily sensitive to local features, rather than the global shape configuration. **Right**: Illustration of our proposed Distorted Shape Testbench (DiST). We hope machine would successfully distinct the images with global shape distortion from the original image, achieving robust visual recognition just like human.

we can see that a stylized trained neural network becomes resistant to the style changes, however, its sensitivity map still shows heavy focus on the **local features** (the eye of the owl in this case), rather than the **global shape** (See results of *Feature Attribution Results* in Figure 1(a), bright pixels in the middle column indicates the area is sensitive to the model perception, we refer the details to Section A.1). Thus, resistance to style transfer does not guarantee the development of network sensitivity to global shapes.

To remedy this problem, we developed a TexForm-based evaluation dataset, called **DiST (Distorted Shapes Testbench**), for evaluating the sensitivity to the global shapes (Figure 1(b)). In this dataset, images were transformed to distort their global shapes while maintaining their texture statistics as well as many of the more local shapes and forms. We used DiST to perform an odd-man-out test on the various models and human subjects to measure their ability to distinguish the original image from its shape-distorted variants as a metric of their global shape sensitivity. We found humans far superior to Style-transfer-trained networks in discriminating the differences in global forms. In fact, the Style-transfer network's performance is no better than the standard CNN that has not been subjected to augmented Style-transfer training. Also, we found that vision transformers (ViTs) fared no better than standard CNN, contradicting the beliefs that ViTs had captured and utilized the global relationship of object parts in this task.

Along with the DiST dataset, we also demonstrate that networks trained with augmented DiST data also do well in discriminating the global shapes of objects if using a carefully designed training approach (we name the method as **DiSTinguish**). It appears that neural networks, in general, tend to take the easiest way out, learning shorter paths and more local features first, and only when they were pressured would they start to acquire more global and structured representations for solving problems. Finally, we found that the DiSTinguish-trained network and Style-transfer-trained network are orthogonal and complementary, as one focuses on global shapes, while the other tends to capture local shapes. Thus, our paper provides a better understanding of the nature of shape bias in networks trained with different augmented datasets and provides a new benchmark for evaluating and enhancing the global shape bias in deep learning models.

## 2 RELATED WORK

Deep Neural Networks (DNNs) have been the cornerstone of the revolution in computer vision, delivering state-of-the-art performance on a wide array of tasks (Luo et al., 2021; Redmon et al.,

2016; He et al., 2016; Brown et al., 2020). However, understanding DNNs have been a vital topic to further advancement of these black box models (Drenkow et al., 2021; Petch et al., 2022; Gilpin et al., 2018). One aspect of understanding DNNs in vision system is identifying the biases they might have when classifying images. Two prominent visual cues are local texture and global shape (Garces et al., 2012; Janner et al., 2017).

Originally, it was believed that DNNs, especially those trained on large datasets like ImageNet, primarily learn shapes rather than textures, as visualization in convolutional neural network shows clear hierachical composition features of various level of object shapes (Zeiler & Fergus, 2014). This belief was also based on the intuitive understanding that shapes are more semantically meaningful than textures for most object categories. However, Geirhos et al. (2018) challenged this belief and showed that DNNs trained on ImageNet have a strong bias towards texture. Our work re-examine their proposed Style-transfer based approach and propose a complementary yet important way to further measure the models' shape bias.

Geirhos et al. (2021) benchmarked various widely used models on the proposed Style-transfer datasets in Geirhos et al. (2018). Among which, networks architectures plays a significant role in improving the models' shape bias. Comparing to the convolutional neural networks (CNNs), newly proposed vision transformer family ViTs (Dosovitskiy et al., 2020) perform significantly better in terms of the Style-transfer based shape bias as well as other corruption based robustness test measured by Paul & Chen (2022). However, we observe that ViTs yield no significant improvement on our proposed global shape distorted test, contrary to both the common belief of ViT's enhanced global relationship modeling ability and the previous shape bias results. Hence our results call for further investigation into methods of improving the shape bias through architecture innovation.

## 3 METHODS

### 3.1 DISTORTED SHAPE TESTBENCH (DIST)

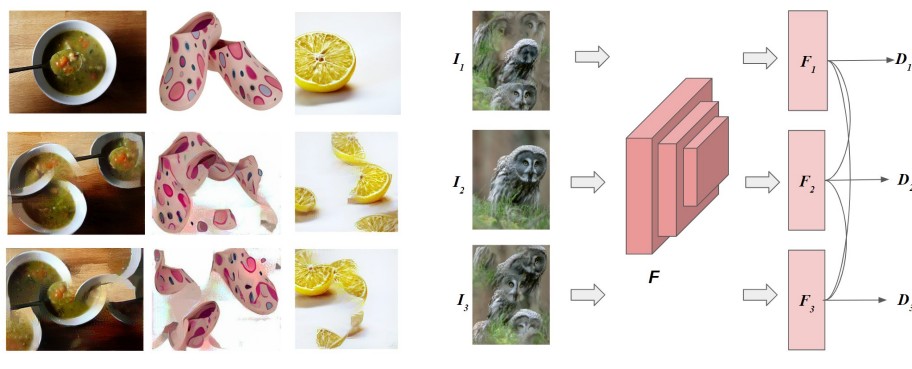

(a) Examples in DiST      (b) The process of DiST for models

Figure 2: Distorted Shape Testbench (DiST)

Our Distorted Shape Testbench (DiST) deliberately compares the representation before and after we apply the global shape distortion. One could imagine there could be many global distortion variants of an original image as the joint spatial configuration of local parts could be arbitrary. To get a quantative measurement of the models' global shape sensitivity, DiST formulates the evaluation metrics as the accuracy of an oddity detection task. The subjects to DiST (machine learning models or human) are asked to select a distinct image from a pool of choices, which consists one original image where the global shape is intact, and $N$ global shape distorted variants of the original image (each of which preserves the local patterns). We pick $N = 2$ for all the DiST test as we observe that increases $N$ will not increase the difficulty of the task.

**Shape Distortion via Texture Synthesis Program** Texture Synthesis allows for the generation of images that retain the original texture details while randomizing the global shape structures. We utilize Gatys et al. (2015) in particular to construct the global shape distorted images for the DiST

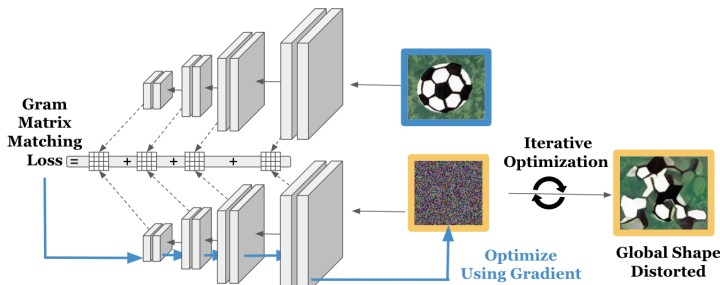

Figure 3: Mechanism of computing global shape distorted images. We implement approach proposed in Gatys et al. (2015). Specifically, we optimize a randomly initialized image (yellow) so that when it passes through a pretrained VGG network, its intermediate layers' gram matrix match the targeted image (blue). This results in preserving the images' local features but randomizing the global structures.

oddity detection task. We illustrate the process of this texture synthesis in Figure 3. For any given target image, $I_t \in R^{(3,H_0,W_0)}$ (blue boundary image in Figure 3), we want to get $I_o \in R^{(3,H_0,W_0)}$ that possess the same local features but distorted global shape (as shown in yellow boundary images on the right). To achieve this, we initialize tensor $I_1 \in R^{(3,H_0,W_0)}$ using value independently sampled from a isotropic Gaussian distribution and complete the process of $I_1 \rightarrow I_2 \rightarrow I_3... \rightarrow I_o$ through minimizing the $L$ as Gram Matrix Matching Loss. Specifically,

$$\frac{\partial Loss}{\partial I_i} = \frac{\partial}{\partial I_i} \sum_l ||\text{Gram}(A_l(I_t)) - \text{Gram}(A_l(I_i))||^2$$

, where $A_l(I) \in R^{(C_l,H_l,W_l)}$ denotes the $l$-th layers activation tensor from which we destroy the global spatial information by computing the channel-wise dot products, i.e. $\text{Gram}(A_l(I_t)) \in R^{(C_l,C_l)}$.

**DiST Metric Formulation** For each trial in the Distorted Shape Testbench (DiST), two shape-distorted versions are generated using distinct random seeds. Each image, denoted as $I_i$, is then passed through the evaluation network $F$ to obtain a feature vector $F(I_i)$ from the final layer. The model identifies the image most dissimilar to the others by calculating the cosine distance between feature vectors. The procedure for this calculation is as follows:

$$D_i = \sum_{j \neq i}(1 - \frac{F(I_i) \cdot F(I_j)}{||F(I_i)||_2||F(I_j)||_2})/N \tag{1}$$

$N$ represent the number of shape-distorted images in each traial, in DiST it would be equal to 2. The dissimilar of the image $I_i$ to other 2 images is calculated as the average of the pairwise cosine distance of each two image pairs. Cosine distance of vector $u$ and $v$ is calculated as $D_C(u,v) = 1 - S_C(u,v)$, where $S_C(u,v)$ is the cosine similarity.

Model will select the images $a$ that is the most different from the other two images based on $D_i$:

$$a = \arg \max_i \frac{exp(D_i)}{\sum_j exp(D_j))} \tag{2}$$

DiST is fundamentally different from the evaluation methods based on styler transfer or other changes of texture details. Those methods apply style transfer operation to generate the evaluation data. The model trained with stylized augmentation could get advantage in those evaluation due to the familiarity of different style domains. In contrast, DiST involves no style transfer operations. Instead, it directly assesses the representations learned by the model to show how sensitive it is due to the change of global structure. This approach eliminates any biases arising from familiarity with stylized images, offering an entirely new angle from which to evaluate shape bias.

### 3.2 Psychophysical Experiments

Human vision is known to exhibit a strong bias towards shape. To quantify the gap between deep learning models and the human visual system, we conducted a psychophysical experiment with human subjects. To align this experiment closely with deep learning evaluations, participants were simply instructed to select the image they found to be "the most different," without receiving any additional hints or context. To mimic the feedforward processes in deep learning models, we displayed stimulus images for a limited time, thereby restricting additional reasoning. Furthermore, participants received no feedback on the correctness of their selections, eliminating the influence of supervised signals.

In each trial, participants were simultaneously presented with three stimulus images for a duration of 800 ms. They then had an additional 1,200 ms, making a total of 2,000 ms, to make their selections. Any response given after the 2,000-millisecond window was considered invalid. To mitigate the effects of fatigue, participants were allowed breaks after completing 100 trials, which consisted of 100 sets of images. We accumulated data from 16,800 trials and 32 human subjects to calculate the overall performance on DiST to represent human visual system. The final results, shown in Fig.5, represent the average performance across all participating human subjects. Further detail and of the psychophysical experiments and how the experiment is conducted can be found at the appendix.

### 3.3 DiSTinguish: Distinguish between the original shape and distorted shape

Deep learning models are excellent learners when we explicitly define them the learning objectives. Stylized augmentation forces the model to learn style-agnostic representation, leading to its impressive performance on stylized domain. Here we would like to directly force the model to distinguish between original shape and distorted. We propose **DiSTinguish**, as shown in Fig.4, a simple supervised training approach to explicitly enforce the constrain to guide the model to learn the global structure of the object. Rather than operating within the confines of an $n$-class classification task, we expand this to $2n$ classes. Where the loss of the network would be: $L(\theta) = -\sum_{i=1}^{2n} y_i \log(p_i)$, where $p_i$ is the predicted probability of the sample belonging to class $i$ and $y$ is the one-hot code for the ground truth label of $2n$ classes.

This expansion incorporates distorted shape versions of each original class as additional, separate classes. In this way we force the model to learn the difference between the original shape class and the distorted one, pushing it to not rely on the local texture detail directly.

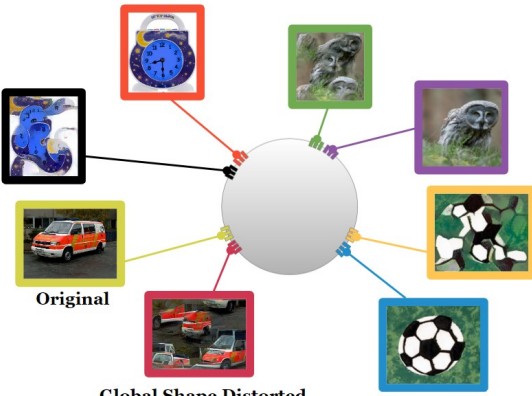

Figure 4: Our DiSTinguish Training Approach, shape-distorted images are added as separated classes

Due to the high time cost of the texture synthesis process (100 optimization steps would take above 55s on a single Tesla V100 GPU, which is the time cost for a single image generation). It's infeasible to generate a full shape-distorted version of the ImageNet1k training dataset. Therefore, we use 10-step optimization results to approximate the 100-step optimization result in DiST, each additional shape-distorted would have 100 images.

During the evaluation phase, the model reverts to $n$-class classification by summing the logits corresponding to the distorted shape and original classes: $z_i' = z_i + z_{i+n}$, where class $i$ and class $i+n$ are the original class and its shape-distorted class, to reduce the output logits $Z$ from $2n$ dimension to $n$ dimension. While for DiST, this remapping is unnecessary, as we directly compare the feature vectors learned by the model rather than the classification logits..

## 4 RESULTS

### 4.1 HUMAN V.S. MODEL'S SHAPE BIAS ON DIST

One of the ideas to evaluate shape bias quantitatively is to use the Cue-Conflict dataset (Geirhos et al. (2021)), where the style of all the images are transferred into the style of another class, causing the conflict between the shape information and texture information. The "shape score" of Cue-Conflict dataset is determined by calculating the proportion of instances where shape, rather than texture, influences the classification, which is defined as Shape Bias Score (Cue-Conflict) = $\frac{\text{Number of Correct Shape Recognitions}}{\text{Number of Correct Recognitions}}$. Models trained on style-transferred images exhibit notably superior performance on this metric. Vision Transformer (ViT) models also outperform traditional CNN architectures, with larger models showing even better results. These findings are based solely on assessments using style-transferred images.

To determine whether the observed trends remain consistent when different evaluation approaches are employed, we re-evaluate the performance of multiple models across different architectures and different training methods using DiST. Those models include: **transformer architectures** (e.g. ViT, BEiT (Bao et al. (2021)), DeiT (Touvron et al. (2022)) and ConvNeXt (Woo et al. (2023))), **traditional CNN architectures** (e.g. ResNet, ResNeXt (Xie et al. (2017)), Inception (Szegedy et al. (2017)), DenseNet (Huang et al. (2017))). **Mobile network** searched by neutral architecture search (e.g. MNasNet (Tan et al. (2019)), MobileNet (Koonce & Koonce (2021))). We also covers model trained with different technique, including **adversarial training**, **sparse activation** (Li et al. (2023)) and **semi-supervised training** (Xie et al. (2020)).

As shown in Fig.5, where both human and model performances are ranked according to Cue-Conflict scores, we observe that several conclusions drawn from Cue-Conflict data do not generalize to the DiST evaluation. Firstly, ResNet50-SIN, which is trained on stylized images (Geirhos et al. (2018)), outperforms other models on the Cue-Conflict dataset but fails to surpass the performance of a normally trained ResNet50 on DiST. This suggests that its high performance on the Cue-Conflict dataset may not be attributed to a better understanding of the global structure but rather to some other learned "short-cut".

Secondly, as shown in Table.1, ViT models do not exhibit a significant advantage over ResNet when evaluated on DiST. Notably, ViT-B even underperforms compared to ResNet50. Additionally, although the performance on Cue-Conflict dataset suggests that large model size leads to higher shape bias, the results on DiST challenge this assumption. A larger model size does not guarantee enhanced capability in perceiving global structures. The performance of various ViT sizes on DiST is incongruent with their parameter sizes; ViT-S outperforms its counterparts, while ViT-B lags behind.

For the human evaluation result, we average human performance data collected from 16,800 trials to obtain the final human performance metric, which is 85.5%, outperforming all the deep learning models. The results consistently show that humans are robust shape-based learners, irrespective of the evaluation method used. In comparison, deep learning models still exhibit a performance gap relative to human capabilities.

Table 1: ResNet50 and different size of ViTs' performance on DiST and Cue-Conflict dataset

| Model | # Param (M) | Cue-Conflict score(%)(↑) | DiST Acc(%) (↑) |
|---|---|---|---|
| ViT-L | 303.3 | **53.8** | 68.5 |
| ViT-B | 85.8 | 43.1 | 54.8 |
| ViT-S | 22.1 | 37.7 | **70.9** |
| ResNet50 | 25.5 | 21.4 | **69.4** |

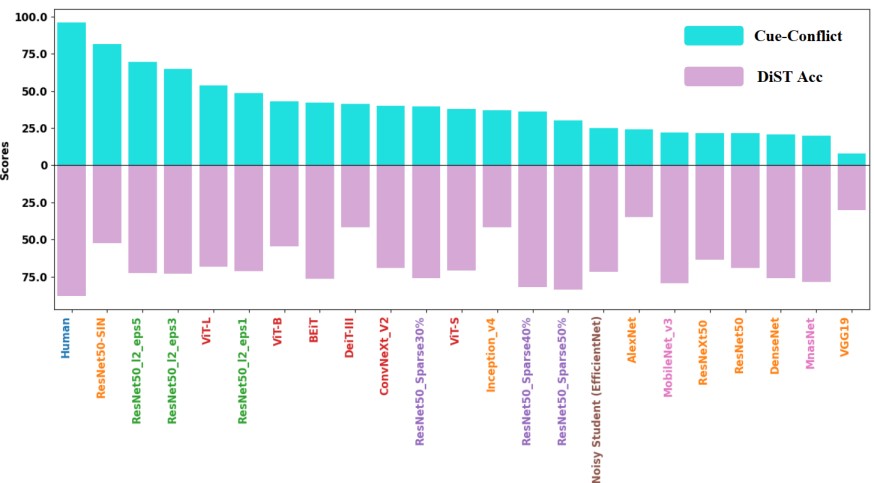

Figure 5: Human and different models' performance on DiST and Cue-Conflict dataset.

## 4.2 ENHANCE SHAPE BIAS BY DISTINGUISH

Our evaluation of current deep learning models using DiST reveals that there is still a performance gap between human and machine capabilities in recognizing significant changes in global shape. To address this, we employ DiSTinguish to explicitly train the model to differentiate between original and distorted shapes. Initial experiments on a small-scale dataset demonstrate the efficacy of our approach. Subsequently, we extend this to the large-scale ImageNet1k dataset. Experimental results indicate that DiSTinguish enables the model to grasp the global shape of objects while maintaining its performance on the original dataset. Notably, DiSTinguish operates orthogonally to Stylized Augmentation techniques, highlighting that these two methods enhance shape perception ability of the model from distinctly different angles.

### 4.2.1 IMAGENET10

For the small-scale experiment, we select 10 classes from ImageNet1K. The choice of the classes is the same as the Imagenette dataset. We train a ResNet50 model from scratch with the same hyperparameter configuration using three different training methods.

1. DiSTinguish-Complete (**DiSTinguish-C**): 20-class supervised learning, the additional classes are the shape-distorted version of the original class, the shape-distorted images are generated using Texture Synthesis with 100-step optimization.
2. DiSTinguish-Approximate (**DiSTinguish-A**): Similar to DiSTinguish-C, while the shape-distorted images are generated using Texture Synthesis with only 10-step optimization.
3. Baseline: Simple 10-class supervised learning without any special augmentation.

We evaluate the above three methods on three different evaluation datasets: **DiST**, style-Transferred version of the evaluation dataset of ImageNet10 (**SIN-10**) and original evaluation dataset of ImageNet10 (**IN-10**). Experiment results on ImageNet10 are shown in Table.2, all the models are ResNet50 trained within 100 epochs. Even though it doesn't reach the same performance of DiSTinguish-C, DiSTinguish-A still surpasses the baseline in both SIN-10 and DiST evaluations. This indicates that DiSTinguish-A serves as an effective approximation, particularly when it is impractical to generate DiSTinguish-C data on large-scale datasets.

### 4.2.2 IMAGENET1K

When scaling up to ImageNet1K, it would be infeasible to generate all the Shape-Distorted version of ImageNet1K train data. Since the experiment on ImageNet10 has shown that using 10-step optimization as an approximation have little influence on the final performance of the model, in the experiments of ImageNet1K, we would use DiSTinguish-A as a replacement for DiSTinguish-C.

Table 2: DiSTinguish on ImageNet10 (Top-1 Accuracy)

|  | IN-10 (↑) | SIN-10 (↑) | DiST (↑) |
|---|---|---|---|
| DiSTinguish-C | 93.2 | 71.6 | 95.5 |
| DiSTinguish-A | 93.2 | 70.0 | 88.4 |
| Baseline | 90.2 | 54.4 | 54.9 |

To demonstrate that DiSTinguish is an orthogonal approach to style transfer methods, we evaluate a ResNet50 model under four distinct training approaches. **(i) Baseline:** The model is trained using pretrained weights without any specialized augmentation. **(ii) Stylized Augmentation:** We employ AdaIN (Huang & Belongie (2017)) to create stylized versions of the ImageNet1K dataset. Each class receives an additional 100 augmented images. **(iii) DiSTinguish:** We use DiSTinguish-A, as employed in our ImageNet10 experiment, as an approximation. An extra 1000 shape-distorted image classes are created, each containing 100 images, and the model is trained as part of a 2000-class classification task. **(iv) DiSTinguish + Stylized Augmentation:** Combining DiSTinguish and Stylized Augmentation together, where the model will trained with 2000-class classification task, and the original 1000 classes images would also contain the stylized images as augmentation.

Except for the pretrained Baseline model, where we directly use the IMAGENET1K_V1 weights, all other models are trained under identical configurations. As shown in the Table.3, DiSTinguish significantly enhances performance on the DiST evaluation while maintaining comparable results on the original dataset. Importantly, as an alternative method for improving shape bias, DiSTinguish is fully compatible with Stylized Augmentation techniques. This compatibility allows for their combined use without any performance degradation in either the stylized domain or the DiST evaluations. This demonstrates that the two methods are orthogonal and can be integrated to comprehensively improve the model's shape bias.

We further examine the feature vectors from the last layer of a ResNet50 model trained under different conditions by visualizing them using t-SNE. As illustrated in Fig.6, both the baseline model and the one trained with Stylized Augmentation fail to effectively distinguish between original and distorted shapes. Their corresponding classes in the feature space show significant overlap. In contrast, the model trained with DiSTinguish clearly separates feature clusters corresponding to original shapes from those of their distorted versions.

Table 3: DiSTinguish and Stylized Augmentation's performance on ImageNet1K

|  | ImageNet1K (↑) | | SIN-1K (↑) | | DiST (↑) |
|---|---|---|---|---|---|
|  | Top-1 | Top-5 | Top-1 | Top-5 |  |
| Baseline | 76.1 | 94.0 | 26.1 | 47.6 | 69.4 |
| Stylized Aug | 78.1 | 94.1 | 52.2 | 75.2 | 73.3 |
| DiSTinguish | 77.7 | 93.8 | 24.9 | 44.7 | 98.6 |
| DiSTinguish + Stylized Aug | 77.8 | 94.0 | **52.2** | **75.7** | **98.7** |

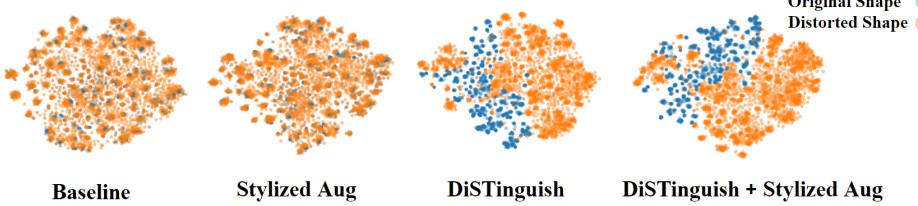

Figure 6: t-SNE visualizations of the feature vector of ResNet50 trained under different approaches.

### 4.2.3 ANALYSIS OF SENSITIVITY MAP

To investigate why Stylized Augmentation fails to significantly improve DiST and how DiSTinguish achieves better performance, we employ SmoothGrad (Smilkov et al. (2017)) to generate sensitivity

maps for a ResNet50 model trained using either DiSTinguish or Stylized Augmentation. Sensitivity maps reveal how responsive the model is to changes in pixel values. To clearly show which regions contribute most to the model's internal representation, we use a binary mask to mask out the pixel that have low sensitivity value.

Across various stylized images, models trained with Stylized Augmentation tend to focus on specific local features that remain relatively invariant to changes in style. For instance, as depicted in the row (ii.), (iii.) and (iv.) in Fig.7, a stylized augmented model may rely heavily on a single eye as the key feature for its decision-making. We hypothesize that the feature associated with the eye remains stable even when the style domain undergoes significant alterations. This enables the stylized augmented model to classify the image correctly despite changes to many texture details.

However, this strategy fail in the DiST evaluation, where the global structure is altered but local features remain constant. Focusing on a specific local region is ineffective for distinguishing a shape-distorted image from the original one, since the features of those local regions are unchanged. As illustrated in the row (i.) of Fig.7, the stylized augmented model fails to account for the global structure, concentrating solely on distinctive features like eyes and neglecting other regions.

In contrast, models trained with DiSTinguish are compelled to make use of global features to effectively differentiate shape-distorted images from original ones. Consequently, the model's sensitive regions are not confined to small, local areas; rather, they extend to larger, global structures. As shown in row (i.) of Fig.7, the model is highly responsive to most parts of the owl, even when they are spatially separated, thus enabling it to perceive changes in the global structure.

This sensitivity to global features persists even in style-transferred images. Compared to stylized augmented models, which fixate on specific local features such as an eye, models trained with DiSTinguish are sensitive to the entire object. This suggests that the two training methodologies engender fundamentally different feature preferences in models. While DiSTinguish encourages models to focus on global structures to discern between distorted and original shapes, Stylized Augmentation prompts the model to rely on features that remain stable across various style domains as a defense against style transfer operations.

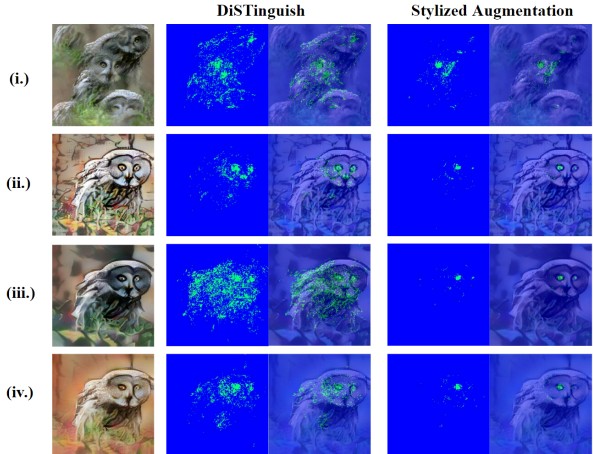

Figure 7: Sensitivity map of ResNet50 trained with DiSTinguish or Stylized Augmentation

## 5 CONCLUSION

In this paper, we introduced the Distorted Shape Testbench (DiST) as a novel metric to evaluate shape bias in deep learning models more effectively than traditional style-transfer-based methods. Our findings reveal three key insights: (i) Existing models acclaimed for shape bias perform poorly on DiST. (ii) ViT models do not outperform CNNs on this new benchmark. (iii) A significant gap remains between human and machine performance on shape recognition. Based on these results, we introduced DiSTinguish, a carefully designed training approach that show significant improvements on DiST without sacrificing accuracy on standard tasks. Our work challenges existing shape bias assessments and provides a new avenue for future research.

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

# A  APPENDIX

## A.1  FEATURE ATTRIBUTION ANALYSIS

The feature attribution analysis is done by using smoothGrad (Smilkov et al. (2017)), one of the gradient-based sensitivity maps (a.k.a sensitivity maps) methods that are commonly used to identify pixels that would strongly influence the decision of the model. Specifically, gradient-based sensitivity maps are trying to visualize the gradient of the class predicted probability function with respect to the input image, which is $M_c(I) = \partial F_c(I)/\partial I$, where $F_c$ is the function that predict the probability that input image $I$ belong to class $c$.

Traditional methods for computing sensitivity maps often suffer from noise, making them difficult to interpret. SmoothGrad improves the quality of these maps by taking the average of the gradients obtained by adding noise to the input multiple times and recalculating the gradient for each noisy version. Specifically, it generate $N$ noisy versions of the input $I$ (e.g. Gaussian noise). For each noisy input, perform a forward and backward pass through the neural network to compute the gradient of the output with respect to each input feature. Then average the gradients across all noisy inputs to create a clear sensitivity map.

During our analysis, to visualize the sensitivity map, sensitivity scores are rescaled to fall within the range of $0$ and $1$. To further clarify the regions of sensitivity, we apply a threshold to create a binary mask based on these scores. In the main experimental context, this threshold is set at $0.15$. Original sensitivity maps without the binary mask will also be presented in the following section for comparison.

## A.2  MODEL TRAINING DETAIL AND CONFIGURATION

**Re-examine the models on DiST**  All the model we used during the evaluation on DiST and cue-conflict dataset are directly from the public pretrained models. ResNet50-SIN is the model trained on only stylized images in Geirhos et al. (2018). For the ResNet50 model we use the IMA-GENET1K_V1 weights from pytorch. Others are the default pretrained weight.

**Experiment on ImageNet10**  In the small-scale experiment of ImageNet10, the class we select are exactly the same as the Imagenette dataset. The class label of the selected class are *n01440764, n02102040, n02979186, n03000684, n03028079 n03394916, n03417042, n03425413, n03445777, n03888257*. The model is trained by using SGD as the optimizer, with learning rate of 0.05, batch size of 256 on a single Tesla V100 GPU for 100 epochs. To eliminate the impact of data augmentation, no special augmentation is appiled during the experiment.

**Experiment on ImageNet1K**  The model we used in the experiment of ImageNet1K, except for the baseline model, which directly using the IMAGENET1K_V1 weights from pytorch, are trained from scratch using ffcv (Leclerc et al. (2023)). All the models are trained for 90 epochs with the start learning rate of 0.1 on a signle V100 GPU. Other configuration remains the same as the default configuration in ffcv for training ResNet50.

## A.3  PSYCHOPHYSICAL EXPERIMENT DETAIL

Psychophysical experiments are conducted using a front-end web application developed in JavaScript. Subjects are instructed to "Find the image that is different from the other two" and can select their answers using keys '1', '2', or '3'. After making a selection, subjects press the spacebar to proceed to the next question.

The trial procedure is illustrated in Fig.8. A set of images appears on the screen after a 300 ms delay and remains visible for 800 ms. In a standard trial, two shape-distorted images and one original image are presented; the correct answer is the original image. Following the 800 ms display period, the images vanish, and subjects have an additional 1200 ms to make their selection, totaling 2 s for decision-making. If no selection is made within this time, the trial is marked as a timeout, and the response is considered invalid. Subjects are given the opportunity to take a break after every 100 images. To prevent the supervision signal, no feedback on answer correctness is provided during the test.

To mitigate the risk of the "oddity pop-out" test devolving into a mere "detection task"—where subjects might focus solely on identifying the original image rather than the one that differs—we incorporate extra catch trials into the experiments, as illustrated in Fig.9.

One catch trial is presented after every 10 standard trials. In each catch trial, two "original images" are displayed: one is a mirrored version of the other, accompanied by a shape-distorted image. It is important to note that there is no overlap between the images used in catch trials and those used in standard trials. In these catch trials, the correct answer is actually the shape-distorted image. The rationale for incorporating such catch trials is to compel subjects to focus on identifying the "different" image rather than the "original" one, thereby aligning the task more closely with how deep learning models behave during DiST evaluation. Results from the catch trials are not included in the final performance metric.

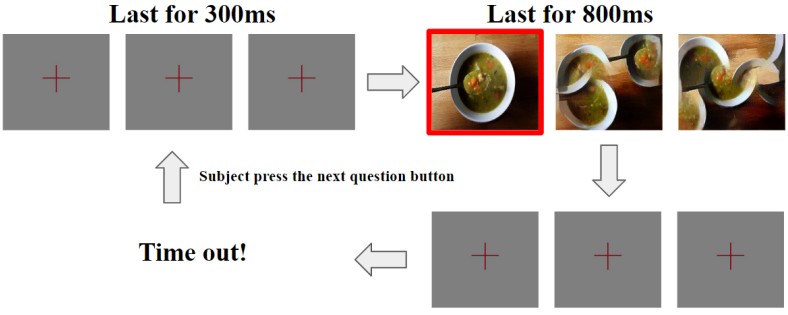

Figure 8: Standard trial in the psychophysical experiment. Image in the red box is the correct answer.

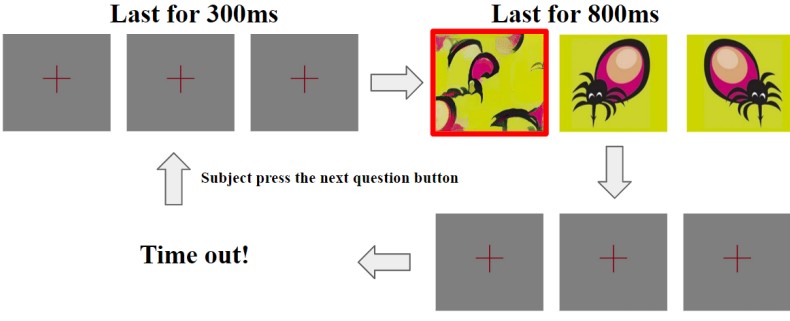

Figure 9: Catch trial in the psychophysical experiment. Image in the red box is the correct answer

### A.4 SENSITIVITY MAP FOR FOUR TRAINING APPROACHES

In this section, we present the original sensitivity maps without binary mask for a ResNet50 model trained using the four different approaches examined in our ImageNet1K experiment. Sensitivity maps serve to illustrate the model's responsiveness to pixel-level changes; a lighter pixel suggests a more significant influence on the model's decision-making process.

As shown in Fig.12, in line with the findings detailed in the main text, the sensitivity map of a model trained with both DiSTinguish and stylized augmentation qualitatively demonstrates the synergistic effect of these methods. Specifically, the model learns to focus on specialized local features that are robust to style transfer, while also becoming attuned to the global structure of the object.

## A.5 BENEFIT OF SENSITIVITY TO GLOBAL SHAPE

Current deep learning model relying on the local texture already shown impressive performance on various tasks. Indeed, using only the local texture is sufficient for many visual tasks. However, under some real-world scenario, having a representation that is sensitive to global shape would have more benefit over the model that focus on the local features. The original evaluation of shape bias through style transferred images is not able to capture such characteristics of the model, which would lead to misunderstanding in related research. In this section, we would use a simple image retrieval task to shows that how does the model focus on the global shape of the image benefit the performance.

The goal to retrieval the original image given the query image, where some part of the object is occluded. The retrieval is simply based on the representation extract from the model. Model would search for the image whose representation is the most similar to the query image according to the cosine similarity. Notice that the original image itself is inside the dataset. The model is either the model trained with style augmentation, believed to have a "shape bias" evaluated by the style transferred images, or the model train with DiSTinguish, which is sensitive to the change of the global shape. As shown in Fig.10, model trained with style augmentation fail to find the original images of those query, while model train with DiSTinguish success.

It's easy to understand that model relying on local feature would fail on this task if the key feature were occluded, and model that focus on the global shape would be robust against partial occlusion. Although as a measurement of "shape bias", evaluated by style transferred images could not provide insight about model's sensitivity to global shape, this would lead to misunderstanding of model's bias, particularly when the real-world applications need such ability as shown in the above example.

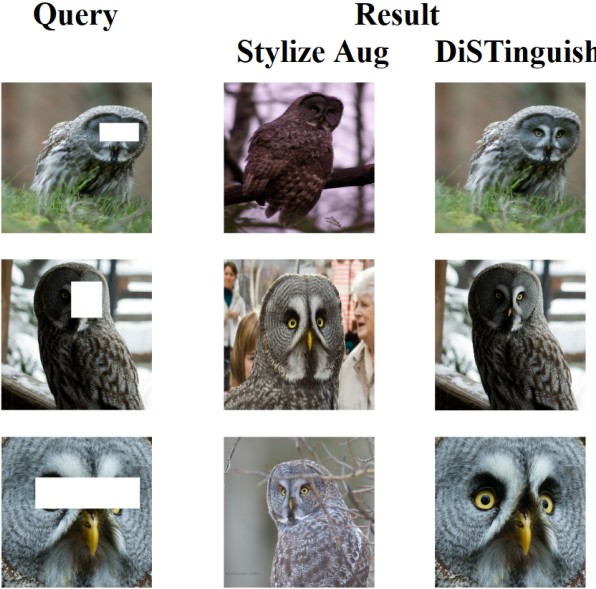

Figure 10: Image retrieval task based on the representation. Although believed to have a "shape bias", model trained with style augmentation fail to find the original images given the partial occluded image. While the model trained with DiSTinguish success on those partial occluded images.

## A.6 MORE EXAMPLE IN DiST

In this section we will show more example in DiST. Fig.11 shows 12 sets of images in DiST, each set consists of one original image (leftmost one) and two generated shape-distorted image. Ideally, we would want the optimization process to generate the image that the local components of the object are disrupted, to test if the model is sensitive to such change of the global shape. But depending on the characteristics of the original images, a small proportion of the generated results can be particularly hard for both humans and models. For example, results in Fig.11 (e) and Fig.11 (k) are the challenging cases, where the objects and background are difficult to distinguish. Those

challenging cases might be less meaningful to evaluate the sensitivity of the global shape of the models. Even the dataset does include some of those cases, most of the content still follow our intent. And human is still able to distinguish most of the images in the dataset (over 85%), which outperform all the model without training with DiSTinguish, showing that such gap about the perception of global shape does exist.

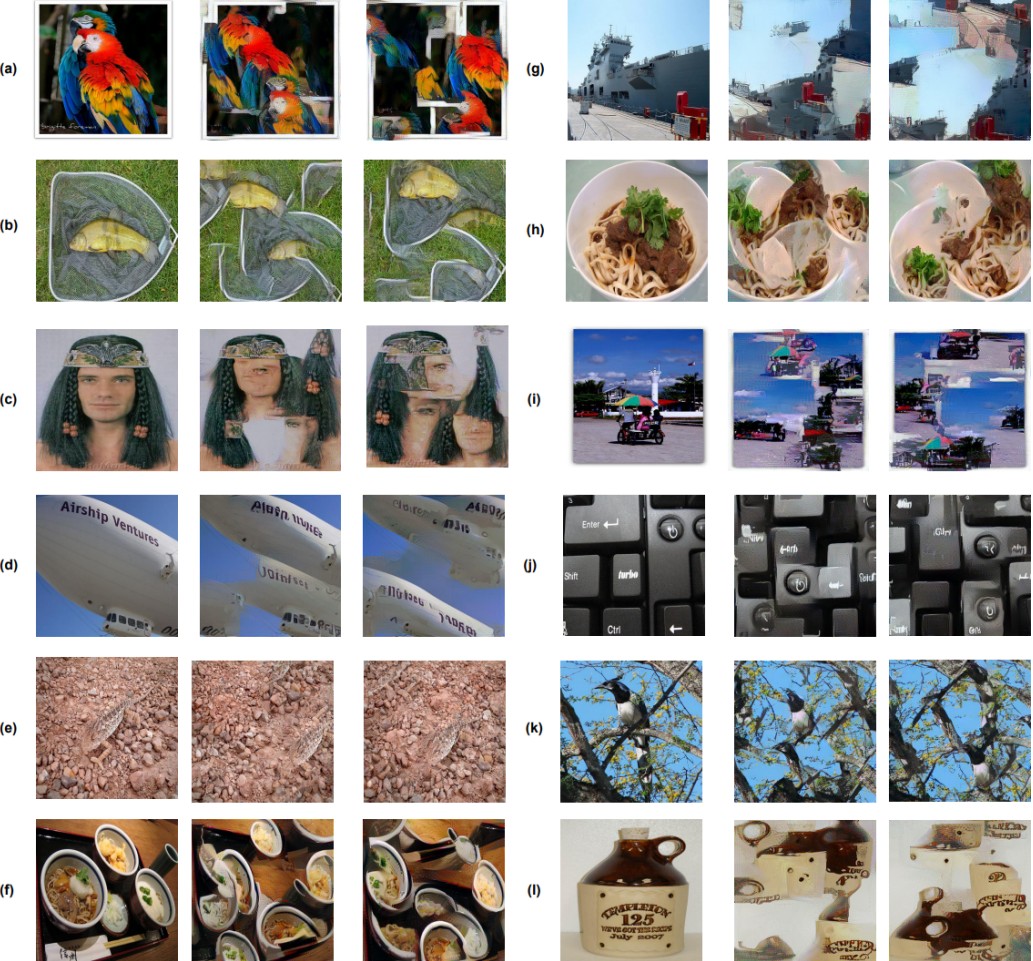

Figure 11: More example of distorted shape images and its original images. The first image in each image set is the original one.

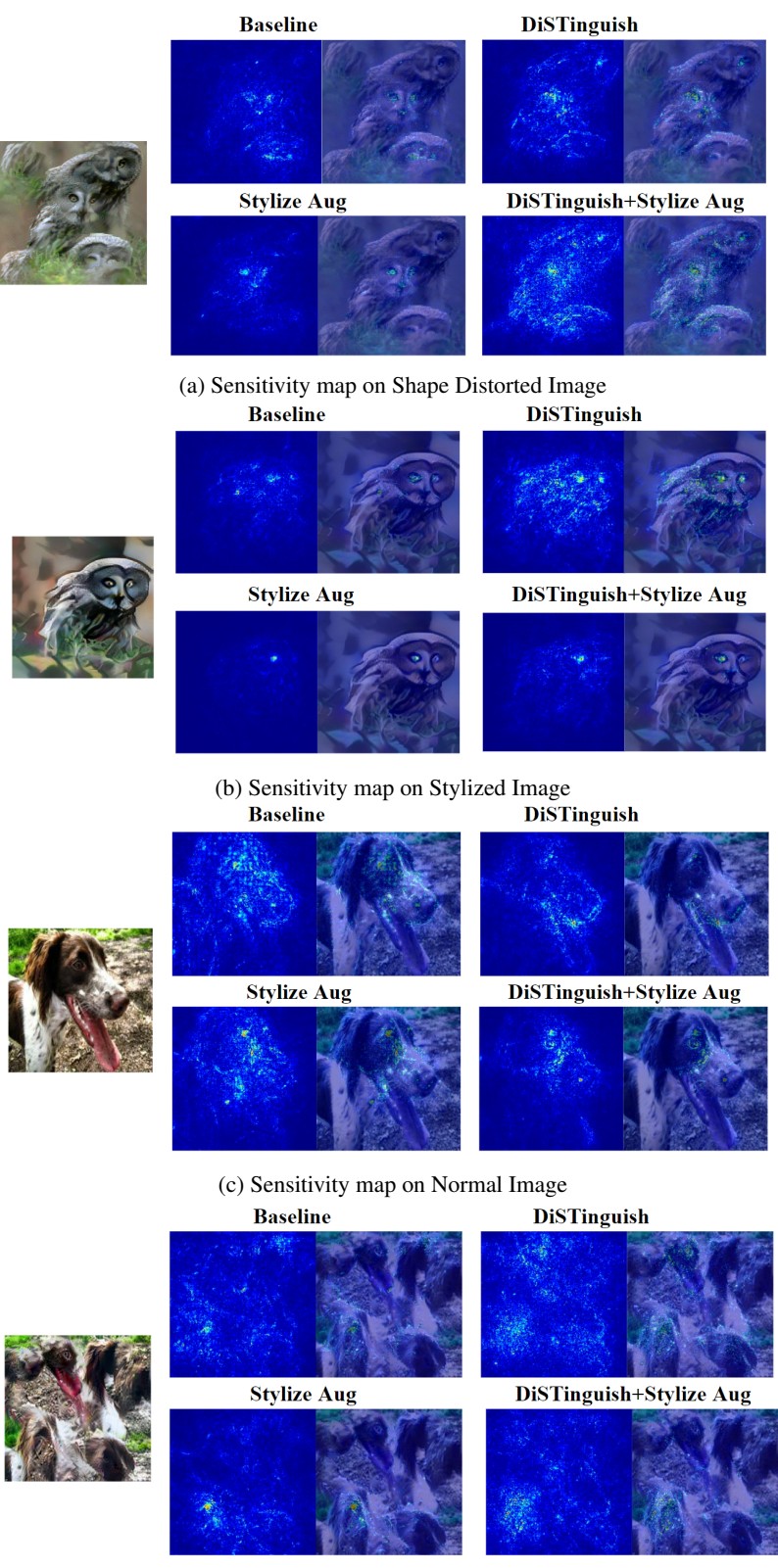

(a) Sensitivity map on Shape Distorted Image

(b) Sensitivity map on Stylized Image

(c) Sensitivity map on Normal Image

(d) Sensitivity map on Shape Distorted Image

Figure 12: Sensitivity map of ResNet50 trained under different methods, the lighter the point is, the stronger that pixel would influence the decision of the model.

