# OpenReview forum: "Does resistance to style-transfer equal Shape Bias? Evaluating shape bias by distorted shape"
_ICLR.cc/2024/Conference — Submitted to ICLR 2024_

### Official Review · Reviewer_1Rgc · 2023-10-17

**Soundness:** 3 good
**Presentation:** 3 good
**Contribution:** 3 good
**Rating:** 6
**Confidence:** 4

**Summary:**

For visual recognition, human rely more on shape while Neural Network rely more on textures, and previous methods trained style transfer augmentation failed to extract global shape information. This paper propose Distorted Shape Testbench as a measurement for global shape sensitivity, and a corresponding training method to improve Visual recognition networks' ability to extract global shape. Qualitative and quantitative experiments have been conducted to show the method's superiority.

**Strengths:**

1. The choice of research topic is insightful, both interesting and practical.
2. The proposed method and benchmark are likely to be helpful helpful to many relevant research fields.
3. The results are promising and the experiments are convincing

**Weaknesses:**

1. In the experiment section, the authors only compared Resnet and ViT, the limited number of network architectures may make the conclusion less persuasive
2. The proposed DiST benchmark is only tested on classification task, while there are many tasks influenced by global shape information. Experiments on different tasks may be needed to show the Versatility of DiST.

**Questions:**

1. The proposed ways of generating shape distortion is based on Neural Style Transfer, does the specific ways of style transfer algorithm used matter? Or could other style transfer/shape distortion methods achieve the same performance?
2. The distorted images have similar textures but different global shape with the originally images. Would contrastive learning help improve the performance in this situation>

---

> ### Author Response · Authors · 2023-11-22
>
> We are glad that the reviewer found the problem insightful and interesting, we think having a clear understanding of “shape bias” would be crucial to the related research. We hope the following feedback can address the reviewer’s questions:
>
>
> **About the limited number of network architectures**: We extend the scope of our benchmarking model, please refer to our general response and section 4.1 and figure 5 in the paper for more details.
>
>
> **About other Style transfer Methods:**
> Two major issues that limit the method we used to generate shape-distorted images is whether they can generate the image which randomly shuffles the local components of the original image, and how much does it take to generate the images. Methods like AdaIN[1] or pix2pix [2] are popular methods to do the style transfer,  however it’s hard for them to do the texture synthesis task where there is no content constraint.  Another practical issue is that many methods using GAN to generate the texture are too slow to produce enough images for either training or testing.
>
> As shown in section 4.2.1, we use an approximate method to generate the shape distorted images, and in the small scale evaluation, it’s as good as the original method. During our experiment, the detailed configuration of generation won’t have a significant influence on the result. Based on our experiment, we think the methods that are practical to be used to generate the image wouldn’t influence the performance.
>
>
> *[1] Huang, X., & Belongie, S. (2017). Arbitrary style transfer in real-time with adaptive instance normalization. In Proceedings of the IEEE international conference on computer vision (pp. 1501-1510).*
>
> *[2] Isola, P., Zhu, J. Y., Zhou, T., & Efros, A. A. (2017). Image-to-image translation with conditional adversarial networks. In Proceedings of the IEEE conference on computer vision and pattern recognition (pp. 1125-1134).*
>
>
> **About contrastive learning:** We did use contrastive learning at the beginning as a possible way of improvement. Compared with traditional contrastive learning methods, we added the shape-distorted variation as a part of the negative pairs to train the model. We performed a small scale test on ResNet18 trained on Imagenette (a 10-class version of ImageNet), the baseline (supervised learning) achieved **39.4%** on DiST and the contrastive learning with additional shape distorted negative pairs achieved **34.3%** on DiST. We think the low performance of contrastive learning is due to the following reasons:
>
> Consider the contrastive loss  (openreview fail to render the whole formulas so we sepeate it):
> $-\sum_{i \in I} \log \frac{\exp ({z}_i \cdot {z}_j / \tau)}{Neg}$
>
> $Neg=\sum_{a \in A(i)} \exp ({z}_i \cdot {z}_a / \tau)$
>
>
> where $j$ is the positive pair for $i$ and for other $a \in A(i)$ are the negative pair. If there are $N$ images in total, there would be $N$ augmented images and $N$ shape-distorted images, meaning that there would be 1 positive pair and $3N-2$ negative pairs for each image. However, the there is only 1 negative pair that would contribute to model performance on DiST, which is the shape-distorted variation of the original image. Even if we don't use contrastive learning, it’s easy for model to distinguish the image with images in other classes and their shape-distorted variations. The only one that would be challenging for the model is the  shape-distorted variation of that image itself. While this negative pair only  occupies 1/(3N-1) of the negative pairs. Even if the model is not optimizing towards the way we want, to separate the representation of the original image with its shape-distorted variations, the contrastive loss can still be optimized well.
>
> We think that the key signal is too weak by using the contrastive loss, and explicitly setting the shape-distorted variation as the other class would provide a strong signal for the model to separate the original image and its shape-distorted variations.

---

### Official Review · Reviewer_dedU · 2023-11-02

**Soundness:** 3 good
**Presentation:** 3 good
**Contribution:** 3 good
**Rating:** 6
**Confidence:** 4

**Summary:**

This paper propose a benchmark dataset, DiST, to measure the global shape sensitivity of models. The images in this dataset is constructed by applying global shape distortion to natural images. Specifically, it follows Gatys et al. (2015), but optimize the intermediate layers' feature but keep the gram matrix not changes. Therefore, the texture information is kept but the shape information lost. Based on this newly constructed dataset, this paper have several interesting observations.

**Strengths:**

This paper is studying an interesting problem, and figured out the missing pieces from the previous studies.

The proposed image construction methods simple but makes sense.

The experimental results further illustrate the value of the proposed dataset.

**Weaknesses:**

The scale of this dataset is too small (less than 10k images IIUC). It would be good to use the proposed method to construct an ImageNet-level dataset.

It would be good to show some well generated images while show some bad images as well, which can help people better understand the pros and cons of the proposed method.

It would be good to benchmark more models. For example, the shape-biased and the texture-biased model from [a].


[a] Li, Yingwei, et al. "Shape-texture debiased neural network training." ICLR 2021

**Questions:**

See weakness. My main concern is the scale of the dataset on both number of images and number of tested models. This paper is very interesting, but potential can explore more.

---

> ### Author Response · Authors · 2023-11-22
>
> We thank the reviewer for the constructive comments and suggestions. We address the reviewers' concerns as follows:
>
> **About the scale of the dataset**: The major concern to limit the scale of the dataset is that we would like to set up a human baseline to compare with the model. If the dataset is too large, it would be difficult to conduct human study based on this dataset. We refer to the dataset scale of *[1][2][3]* where they benchmarked both humans and models on various Out-Of-Distribution datasets, as each dataset has around 1280 images. As our dataset has 7200 (2400*3) images, although it’s not yet a ImageNet-level dataset, we think it would be ample to evaluate global shape sensitivity of the model.
>
>
> **About more examples of the generated data**: We have updated more examples of DiST in the appendix. While it’s hard to give a definition about “good” images and “bad” images, we think the generated result that shuffles local components of the object is the ideal case we want to evaluate the model (like the owl example in the paper). There do exist a small proposition of  examples that are difficult for both human and model, where the background is complex and hard to be separated from the object, the structure (global shape) of the image is hard to be captured. Fig 10(e) is such a case. Please check section A.6 for the details.
>
>
> **About benchmarking more models:** As mentioned in the general response, we have added additional experiments to evaluate the performance to cover various architectures and training methods, to show how they influence the model’s performance on both DiST and cue-conflict dataset. Although we would like to test the model mentioned in your reference, the pertained model in their git repo seems to be broken and we can’t perform the evaluations on them. Please check the section 4.1 and figure 5 in the paper for more details.
>
>
> *[1] Geirhos, R., Temme, C. R., Rauber, J., Schütt, H. H., Bethge, M., & Wichmann, F. A. (2018). Generalisation in humans and deep neural networks. Advances in neural information processing systems, 31.*
>
> *[2] Geirhos, R., Narayanappa, K., Mitzkus, B., Thieringer, T., Bethge, M., Wichmann, F. A., & Brendel, W. (2021). Partial success in closing the gap between human and machine vision. Advances in Neural Information Processing Systems, 34, 23885-23899.*
>
> *[3] Geirhos, R., Rubisch, P., Michaelis, C., Bethge, M., Wichmann, F. A., & Brendel, W. (2018). ImageNet-trained CNNs are biased towards texture; increasing shape bias improves accuracy and robustness. arXiv preprint arXiv:1811.12231.*

---

### Official Review · Reviewer_PsLQ · 2023-11-02

**Soundness:** 2 fair
**Presentation:** 2 fair
**Contribution:** 1 poor
**Rating:** 3
**Confidence:** 4

**Summary:**

The paper is centralled concerned with robustness of recognition to shape distortion. The question asked in "does resistance to style transfer equal shape bias?" By which the authors seem to mean that recogntion algorithms that have been extended to be decently robust to texture change in objects are not robust to shape changes.

The paper introduces a data set dseigned for the problem, and a distance they call DiST.

**Strengths:**

I like the odd-one-out test, which I think is a sound way to measure subjective distances.

**Weaknesses:**

It is clear from simple observation that texture and shape can both change, and that neural nets are currently configured to rely primarily on texture. So the paper is not saying too much in that regard.

The dataset introduced comprises an unconvincing collection of images. These include images that have been assumbled from sub-images of the object at different scales and points of view, peeled fruit (and peeled footballs). The reason this is not convncing is that the shape changes within a class will rarely if ever be expressed in such a way. Far more common would be simple geometric distortions. As an example, a "man" could be distorted into a "strong man" by increasing body and muscle size compared to the head. A second reason is that in some cases at least shapes can change in very unexpected ways but still be reognisable. Dali's melting watches are an example.

The fact that current models do not perform so well on the new dataset is not surprising. It is not so hard to contruct such a dataset. And the conclusion that humans out perform machines is equally unsurprising.

**Questions:**

Why not use some kind of warping to resist changes in shape? (Inter-class warps should be distinct from intra-class warps).

What is your justification for using images made of pieces of photos of an object? (These are not shape distorted, they are mosaics of regualarly shaped parts)

People can draw things - say a face - in all sorts of shapes. Eyes can be round, lines, crosses, diamonds and many more shapes.
Why not include such examples in your dataset (these examples are real and common in art)

---

> ### Author Response · Authors · 2023-11-22
>
> We thank the reviewer for the constructive criticism and feedback. We hope the clarification below addresses your concerns. And we hope the review can check overall response 1 where we gave a clear clarification of concept and the contribution of the paper.
>
> **neural nets are currently configured to rely primarily on texture**:
> It has been discovered years ago that **normally trained** neural networks are biased towards texture. While our paper wants to emphasize another crucial aspect: the current way to evaluate the shape bias of the model can only show how robust it is against the change of the style, models that achieve high performance on this evaluation are believed to have a “shape bias”. However, as shown in Figure 1 left, even **the model with “shape bias”** through style augmentation, still heavily focuses on the local feature. According to our knowledge, few research in deep learning have paid attention to the definition of “shape bias” but simply evaluate through those style-transferred images. This could cause some confusion when trying to understand the bias of the model. We want to use DiST to show how the model is sensitive to the global shape of the image,  which is missing in the previous metrics. And it would benefit the model in particular application (as shown in general response 3).
>
> **Why not use some kind of warping to resist changes in shape? (Inter-class warps should be distinct from intra-class warps). What is your justification for using images made of pieces of photos of an object? (These are not shape distorted, they are mosaics of regularly shaped parts)**
>
> It appears there may be a slight misunderstanding regarding the concept of "shape distortion" as used in our study. Our purpose of using texture synthesis is to create an image that has the same texture details as the original image, but the arrangement of local components in the image is disrupted, leading to a different global shape.  Like the examples shown in figure 2, the components of the owl are separated and shuffled, even if the local component would remain relatively the same (like the eyes), it would be hard to regard it as an owl due to its weird arrangement. That’s also the reason why styled augmented models still fail on this task, since it is not aware of the global shape, but only focuses on the eye of the owl, which is the same in the generated result.
>
> **People can draw things - say a face - in all sorts of shapes. Eyes can be round, lines, crosses, diamonds and many more shapes. Why not include such examples in your dataset (these examples are real and common in art)**
>
> The term “shape” we used in the paper is more related to the global arrangement of local components, which we called “global shape”. Although the small local component can be drawn using different shapes (circle, diamonds), this does not represent the “global shape” of the object. And it might still be within the scope of “texture” if we consider the model that heavily focuses on the local texture. For example, changing a round eye into a square eye, but not changing the position of the eye, would not influence the global shape, and can be regarded as a change of local texture. Since we know that models, even with style augmentation, still focus on those local textures, we want to make sure that the image in our test has exactly the same texture as the original images, the scope of this “texture” might also cover what you consider as the shape. Given those considerations, using human drawing might not be a very good option because it’s hard for us to control those texture details. Otherwise we are not able to ensure that model can’t distinguish between those images is due to different local texture, or different global shape (i.e. arrangement of local component )

---

### Official Review · Reviewer_4QYv · 2023-11-05

**Soundness:** 3 good
**Presentation:** 3 good
**Contribution:** 3 good
**Rating:** 6
**Confidence:** 4

**Summary:**

This paper introduces a new benchmark to measure how well models can handle distorted shapes. They show that removing texture alone is not sufficient to make models robust to shape distortions. They also suggest that combining their method with Stylized Aug training can improve the models’ robustness to both style and shape variations.

**Strengths:**

This paper is clear and well-structured. The authors share their code, which facilitates the replication of the experiments. The benchmark they propose is original, as it focuses on shape distortions rather than style variations. They also provide a human baseline for comparison.

**Weaknesses:**

I have some doubts about the need for models to be robust to shape distortions. The examples in Fig. 2(a) seem very challenging even for humans. Are there any real-world applications that require such ability? The distorted shapes do not seem to have any physical meaning.

And what is the benefit of using global shape features if the model can already classify the image based on local shape features?

I also noticed that DiSTinguish itself worsens the performance on SIN-1K, as shown in Table 3. Can you explain why this happens?

**Questions:**

Please check weaknesses.

---

> ### Author Response · Authors · 2023-11-22
>
> Thank you for your time and feedback. Below we address your concerns.
>
> “**need for models to be robust to shape distortions**”
>
> We use DiST to reflect the sensitivity of the model to such change of the global shape, and how much it relies on the local texture. High performance in the DiST indicates more sensitivity to the global shape and less reliance on the local texture. The goal of DiSTinguish is not to make the model robust against shape distortion, actually the opposite, to force the model to be sensitive to shape distortion.
>
> If a model claims to have a real “shape bias”, it should be sensitive to the global shape of the object. As we shown in figure 1 left, the model claimed to have  “shape bias” is actually still relying on the local texture.  Although style augmentation is useful against the change of style, we want to highlight that it shouldn't  be regarded as “shape bias”, since the representation of the model is highly relying on the local texture. We think that clarifying this would be useful for the future research to escape some confusion when trying to understand the bias of the model, since the “shape bias” reflected by stylized image can only show the resistance to the change of texture, rather than sensitivity to glocal shape.
>
> **Benefit of Shape Bias / Sensitivity ot the change of global shape / Less reliance of on the local texture in real-world scenario:**
>
> For the model that is sensitive to change of global shape, which directly reflects the shape bias, its representation would be robust even if some local texture has changed. To illustrate this, we use the representation learned by DiSTingiush  and Style augmentation to do image retrievals under partial occlusion conditions. Model trained with DiSTingiush is able to find the original image even if some part is occluded, while the stylized augmented model fails to find the original image due to its reliance on particular local features. Please check the section A.6 of the appendix for the details.
>
> **“The distorted shapes do not seem to have any physical meaning.”**
>
> Although shape distortion is an artifact that rarely has any physical meaning, the idea of using shape distortion is that we want to directly test to what extent the model is relying on the local texture rather than the global shape.
>
> **“DiSTinguish itself worsens the performance on SIN-1K”:**
>
> In table 3 of the paper: model trained with DiSTinguish is 1.2% worse than the baseline. We don’t think it’s a significant drop in performance. We divided the SIN-1K in to 50 subsets and calculate the mean and standard deviation, the perform of DiSTinguish in SIN-1k with error bar would be 24.86%  $\pm$ 1.51% and the baseline would be 26.08% $\pm$ 1.30%. And during our experiment, the performance of models on stylized-images depends on what styles the images transfer to. If the dataset uses a different set of styles to transfer, the evaluation result would also be slightly different. Given the above considerations, that small drop on the performance is not very significant.

---

### Author Response · Authors · 2023-11-22
**Overall response 2: Additional experiment and revised draft**

This is the follow-up comment for the overall response, we have add more experiment according to the reviewers' feedback. Changes and additional experiments are summarized below:

2. **Other tasks that would benefit from strong global shape bias** We use a simple image retrieval task to show the real world applications when the sensitivity to global shape would bring benefit to the model.

Although believed to have “shape bias”, models trained with stylized augmentation would fail to find the original image given the image with partial occlusion, while the model trained with DiSTinguish success.

We want to use this experiment to show that although as a measurement of “shape bias”, evaluating through style transferred images could not provide insight about model’s sensitivity to global shape, this would lead to misunderstanding of model’s bias, particularly when the real-world applications need such ability as shown in the above example. This part has been updated in the section A.6 of the appendix.

3. **Benchmarking more models**: We extended the scope of the benchmarking. We compared the performance of cue-conflict score and DiST accuracy across multiple transformer-based architectures (**ViT, BEiT, DeiT, ConvNeXT**) , traditional CNN architectures (**ResNet50, ResNeXt, DenseNet, Inception**), mobile model found by neural architecture search (**MNasNet, MobileNet**). We also test the same model trained with different techniques, including **adversarial training, sparse activation, and semi-supervised learning**. This part has been updated in section 4.1 in the paper, figure 5 has also been updated.

4. **More examples of DiST**: We provide more examples in DiST at the appendix part in the paper. Check section A.6 for the details.


*[1] Geirhos, R., Rubisch, P., Michaelis, C., Bethge, M., Wichmann, F. A., & Brendel, W. (2018). ImageNet-trained CNNs are biased towards texture; increasing shape bias improves accuracy and robustness. arXiv preprint arXiv:1811.12231.*

*[2] Brendel, W., & Bethge, M. (2019). Approximating cnns with bag-of-local-features models works surprisingly well on imagenet. arXiv preprint arXiv:1904.00760.*

*[3] Zhang, H., Goodfellow, I., Metaxas, D., & Odena, A. (2019, May). Self-attention generative adversarial networks. In International conference on machine learning (pp. 7354-7363). PMLR.*

*[4] Ayzenberg, V., & Behrmann, M. (2022). The dorsal visual pathway represents object-centered spatial relations for object recognition. Journal of Neuroscience, 42(23), 4693-4710.*

*[5] Wang, Xiaolong, et al. "Non-local neural networks." Proceedings of the IEEE conference on computer vision and pattern recognition. 2018.*

*[6] Li, Tianqin, et al. "Emergence of Shape Bias in Convolutional Neural Networks through Activation Sparsity." NeurIPS 2023.*

---

### Author Response · Authors · 2023-11-22
**Overall response 1: clarification of global shape sensitivity**

We thank all the reviewers for their insightful comments and constructive feedback. Overall the reviewers think our paper is “original”, “sound”,  “interesting”, and have overall good contributions. There are several highlights pointed out by the reviewers and we would like to address it here:

1. **Clarification of global shape sensitivity**: We want to emphasise that “shape bias” consist of two aspects: one is that image perception should not rely on the image styles or simple statistics but rather robust to various domains changes, thus achieving domain generalization [1]; the second is that the model should not rely on the simple local patterns but rather taking into the consideration of the global arrangement of the local parts. This is highlighted in the BagNet [2], which shows CNNs only use localized patterns to perform recognition. Human study, on the other hand, suggests our vision system utilizes the global arrangement.

Our paper provides a benchmark to evaluate the second aspect of the “shape bias” and propose an effective approach to solve it. Below are some further insights our paper contributes to the fields:

(1) Incorporating the global configuration of an image into deep neural networks’ representation learning is a long desired property for the vision field. The idea of hierarchical recognition advocates that the image scenes are composed of objects, where objects are composed of parts, recursively down to the very basic local features like edges and corners. This recursive hierarchical perspective is directly reflected in the hierarchical design of CNNs, hoping that the higher layer neurons to recognize the larger semantic global patterns composing the low level local primitive features. However, learning the global arrangement relationship in such hierarchical network architecture is not easy to enforce, requiring intricate coordination across layers. Utilizing local features in CNN might be enough as BagNet suggests for object recognition, however, other tasks utilizing CNNs such as image synthesis show significant deficits if the global relationship are not properly configured (e.g. few-shot images learn clear local parts easily but struggle at the proper arrangement of these local parts [6]). Part of the reason is that global arrangement has less repeating data whereas local features are abundant in images.

To learn the global component arrangement, non-local / self-attention mechanisms were introduced in CNNs [5]. SAGAN [3] shows the benefit of explicitly modelling global configuration, leading to synthesizing images with coherent global structures. The rapid shift towards ViTs in perception from traditional CNNs also suggests the large desire of explicitly incorporating the global arrrangement patterns. However, these arrangement patterns are far more varying and less repeating in the normal images than the local features, therefore the model has no desire to utilize it to make prediction despite the Self-Attention's mechanism design. Our results confirm that the improvement of ViT over CNN are limited in terms of global arrangement sensitivities. Our method, on the other hand, constructs a synthetic dataset, taking an alternative route to force model to incorporate global component arrangement into its representation. We show this is effective approach to allow the model achieve human performance in terms of global shape sensitivity.

(2) This paper provides insights to neuroscience too. Global relationships contribute a lot to human perceptions. A simple thought experiment would show this: Imagine an image where the location of your best friend’s eye, nose, and mouse getting randomly permuted, how would you feel when looking at that image? It would look anything but a person’s face and invoke terrifying feelings despite the local parts being perfectly preserved. This is quantitatively reflected in our test too where humans perform much higher than the existing models on global pattern sensitivities. What causes this phenomenon? [4] suggests human vision achieves this via involving the dorsal stream to sense the global organization instead of purely based on the traditional belief of ventral stream. Given that the CNN was primarily motivated to represent the ventral stream, [4]’s suggestions pounder the vision community with the question from the neuroscience perspective: Is the hierarchical architecture of CNNs alone sufficient for modeling our vision systems? Or our brain also implements expensive relationship models like the transformers for modeling global components arrangements. Our paper advocates for the former. We show that CNNs alone can be fine tuned to achieve this long desired global relationship modeling, with the help of a particularly constructed dataset.

(3) We show that DiSTinguish works orthogonally with the Style-transferred augmentation, making two methods together a working solution for achieving full meaning of “shape bias”.

---

### Meta-Review · Area_Chair_hKg1 · 2023-12-09

**Metareview:**

The paper introduces a new evaluation method, DiST, to assess the global shape sensitivity of deep learning models, revealing that models trained with style-transfer images develop a bias towards local shapes rather than global ones and that training with DiST images can improve model performance in recognizing global shapes while maintaining accuracy in standard image classification tasks. There are 3 borderline acceptances and 1 rejection from the reviewers for this paper. The reviewers have generally shown concerns on the experimental results, and the scale of the experiments, and there is also an argument that the re-arranging pieces of an image does not necessarily represent shape change. After carefully reading the paper, the rebuttal and the reviews, the AC agrees that the concerns are valid and recommends rejection.

**Justification For Why Not Higher Score:**

The reviewers have generally shown concerns on the experimental results, and the scale of the experiments, and there is also an argument that the re-arranging pieces of an image does not necessarily represent shape change.

**Justification For Why Not Lower Score:**

N/A.

---

### Decision · Program_Chairs · 2024-01-16

Reject